# Nanocomposite Foams with Balanced Mechanical Properties and Energy Return from EVA and CNT for the Midsole of Sports Footwear Application

**DOI:** 10.3390/polym15040948

**Published:** 2023-02-14

**Authors:** Boon Peng Chang, Aleksandr Kashcheev, Andrei Veksha, Grzegorz Lisak, Ronn Goei, Kah Fai Leong, Alfred ling Yoong Tok, Vitali Lipik

**Affiliations:** 1School of Materials Science & Engineering, Nanyang Technological University, 50 Nanyang Avenue, Singapore 639798, Singapore; 2Sportmaster Innovation Centre, 3 Cleantech Loop, #06-14 Cleantech 2, Singapore 637143, Singapore; 3Residues and Resource Reclamation Centre (R3C), Nanyang Environment and Water Research Institute, Nanyang Technological University, 1 Cleantech Loop, Clean Tech One, Singapore 637141, Singapore; 4School of Civil & Environmental Engineering, Nanyang Technological University, 50 Nanyang Avenue, Singapore 639798, Singapore; 5School of Mechanical & Aerospace Engineering, Nanyang Technological University, 50 Nanyang Avenue, Singapore 639798, Singapore

**Keywords:** ethylene-vinyl acetate, multi-walled carbon nanotubes, foams, plastic upcycling, dynamic impact response, energy return

## Abstract

Polymer foam that provides good support with high energy return (low energy loss) is desirable for sport footwear to improve running performance. Ethylene-vinyl acetate copolymer (EVA) foam is commonly used in the midsole of running shoes. However, EVA foam exhibits low mechanical properties. Conventional mineral fillers are usually employed to improve EVA’s mechanical performance, but the energy return is sacrificed. Here, we produced nanocomposite foams from EVA and multi-walled carbon nanotubes (CNT) using a chemical foaming process. Two kinds of CNT derived from the upcycling of commodity plastics were prepared through a catalytic chemical vapor deposition process and used as reinforcing and nucleating agents. Our results show that EVA foam incorporated with oxygenated CNT (O-CNT) demonstrated a more pronounced improvement of physical, mechanical, and dynamic impact response properties than acid-purified CNT (A-CNT). When CNT with weight percentage as low as 0.5 wt% was added to the nanocomposites, the physical properties, abrasion resistance, compressive strength, dynamic stiffness, and rebound performance of the EVA foams were improved significantly. Unlike the conventional EVA formulation filled with talc mineral fillers, the incorporation of CNT does not compromise the energy return of the EVA foam. From the long-cycle dynamic fatigue test, the CNT/EVA foam displays greater properties retention as compared to the talc/EVA foam. This work demonstrates a good balanced of mechanical-energy return properties of EVA nanocomposite foam with very low CNT content, which presents promising opportunities for lightweight–high rebound midsoles for running shoes.

## 1. Introduction

Ethylene-vinyl acetate copolymer (EVA) foam is extensively used as cushioning material for floor mats, sports components, and midsole of footwear applications. This is due to its lightweight, great surface resilience, low energy absorption, and high energy return. Foot orthoses made from EVA show improved running economy and comfort of athletes as compared to other polymer foams [1]. However, foamed EVA is relatively soft and lacks mechanical properties. The development of comfort and high mechanical performance EVA foam midsoles through various modifications is an important aspect of the shoe manufacturing industry. In the mass production of EVA foam, mineral additives such as talc and calcium carbonate are usually incorporated in EVA formulation to improve the foam’s overall mechanical properties and to reduce cost. Despite the mechanical property’s enhancement, the traditional practice of using these mineral fillers as a reinforcing agent usually comes with a trade-off of the foam’s dynamic energy return performance and resilience. These properties are critical for the shoe’s midsole rebound and energy return to increase athlete running performance. For instance, it was found that the incorporation of calcium carbonate (CaCO_3_) in EVA/PU foams deteriorated the foam’s performance in terms of resilience, abrasion resistance, and compression set [2].

Compared to the conventional composite foams, nanocomposite foams produced from polymeric matrix with the incorporation of nanoparticles have gained much attention in recent years [3]. Nanoparticles are well known for their effective large surface area (better filler-matrix interactions) which could provide an effective reinforcement effect for polymers. In addition, nanoparticles may possibly act as a strong nucleating agent during the bubble formation in the polymer foams processing. This will result in the smaller cell size and higher cell density of the polymer foams. The surface chemistry, size, concentration, and shape of the nanoparticles are strongly affecting the properties and microcellular structures of the polymeric foams. Furthermore, the rheological properties of the polymer change significantly with the presence of a nano-fillers network in the matrix. For instance, a nano-sized fillers such as organoclay with the presence of a compatibilizer can increase the viscosity and prevent cell coalescence, which results in the higher cell density of the foam [4]. Qewami et al. [5] reported a significant improvement in the viscoelastic properties of PE/EVA blends after the addition of expanded graphite nanoplatelets, which prevented foam cell wall rupture and gas release during the foaming process.

Carbon nanotubes (CNT) are one of the enticing nanomaterials widely used in polymer composites advancement [6]. This is due to the incorporation of CNT in polymers that not only elevates the mechanical properties of the polymer but could also enable additional functional properties such as improved thermal and electrical conductivity, electromagnetic interference shielding, sound absorption, sensing property, and shape memory effects. In addition, only a small amount is required for remarkable improvement. Polymeric foams incorporated with different types of CNTs prepared by both chemical and physical foaming have been reported in numerous studies. These include polyurethane (PU) [7,8,9], poly(methyl methacrylate) (PMMA) [10,11,12], ethylene-propylene-diene-monomer (EPDM) [13], EVA [14], polycaprolactone [15], etc. Various properties improvement on the EVA was reported after being reinforced with CNT [16,17,18,19]. Park and Kim [14] reported that the mechanical properties and elastic recovery of the EVA foam were improved significantly, while the surface resistivity was reduced after the incorporation of CNT.

However, most of the polymer filled with CNT composite foam studies thus far are not focused on the dynamic impulse rebound (energy return) and the dynamic response of the foam, particularly for EVA foam used for sportswear midsole application. In our previous works, we found that carbon-based nanomaterials including carbon black, graphene, and CNTs have properties desirable in developing high-performance foam for sport footwear soles when used as reinforcing fillers [20]. EVA filled with a very low content of graphene shows a striking improvement of approximately 30% increase of stiffness, superior abrasion resistance, and better-cushioning property as compared to the reference sample, which is the desire for running shoes in terms of energy repulsion [21].

In this work, we investigate the physical, mechanical, dynamic impact properties and dynamic fatigue performance of EVA incorporated with different kinds of CNTs derived from upcycling of commodity plastics. All the produced foams are subjected to various testing that specifically imitates the running conditions of the athlete. Furthermore, the physical and mechanical properties, cell morphology, as well as energy return of the developed CNT/EVA foams were compared to the conventional talc/EVA foams that are widely used for making the midsole of running shoes.

## 2. Materials and Methods

### 2.1. Materials

The ethylene-vinyl acetate copolymer (EVA) containing 27.5 wt% of vinyl acetate content with a melt-flow index of 5.5 g/10 min (grade UL00628) was purchased from Zhonghua Quanzhou Petrochemical Co., Ltd., Fujian, China. Dicumyl peroxide (DCP) was purchased from Sigma-Aldrich. All other ingredients are industrial grades chemicals, i.e., zinc oxide (ZnO), stearic acid (SA), talc, and azodicarbonamide (ADC) purchased from Shanghai Macklin Biochemical Co., Ltd., Shanghai, China.

In order to achieve better compatibility between the CNT and EVA, two variants of multi-walled CNT were used (supplied by Nanomatics Pte. Ltd., Singapore). Multi-walled CNT were prepared by upcycling polyolefin plastics. A mixture of low-density polyethylene, high-density polyethylene, and polypropylene was used as a feedstock. Plastics were first pyrolyzed to generate oil and non-condensable pyrolysis gas. After the separation of oil by the condensation process, the gas was used as a precursor for the synthesis of the CNT via a catalytic chemical vapor deposition process. To purify and functionalize CNT, two methods were used. (1) Oxygenated CNT (O-CNT) were prepared using chlorination above 1000 °C with a modified method from [22] and subsequent treatment with air as described in [23]. (2) Acid-purified CNT (A-CNT) were prepared by boiling CNTs in the mixture of deionized water and 70% nitric acid (4:1 volume ratio) followed by filtration and drying at 110 °C.

### 2.2. Characterization of CNT

N_2_ adsorption and desorption isotherms of CNT were collected at −196 °C (Quantachrome Autosorb-1 Analyzer). Specific surface area was calculated using the Brunauer–Emmett–Teller (BET) model and total pore volume was determined from N_2_ uptake at a relative pressure of 0.95–0.96. Graphitization degree was calculated from the areas of D- and G-bands of carbon in Raman spectra acquired at a laser wavelength of 532 nm (XploRA PLUS, Horiba Scientific). X-ray photoelectron spectroscopy (XPS) of CNTs was conducted using an XPS Shimadzu Kratos AXIS Supra. The morphology of CNTs was characterized by a field emission scanning electron microscopy (FESEM, JEOL JSM-7600F). Ash content was determined by the combustion of samples in the air at 900 °C for 2 h.

### 2.3. EVA Foam Preparation

EVA and all the foaming auxiliaries, i.e., SA (processing aid, 0.5 wt%), ADC (blowing agent, 2.5 wt%), ZnO (activator to lower the decomposition temperature of the ADC, 1.5 wt%), DCP (crosslinking agent, 1 wt%), CNT (reinforcing and nucleating agent), and talc (reinforcing and nucleating agent), were sequentially mixed in a two-roll mill, (model XH-401CEW-160 from XIHUA Testing Machine Co., Ltd. Guangdong, China) at 70 °C. DCP was added at the end of the mixing to avoid premature crosslinking of the mixture. The compound was discharged after all the ingredients were mixed uniformly within the EVA sheet (approximately ~20–25 min of high-speed mixing time to achieve uniform color and texture of the sheet). The rolled EVA sheets were stored in the laboratory environment for 24 h before subsequent processing.

The foaming of the EVA was carried out in two steps of compression stages. The rolled EVA sheet was first cut into appropriate sizes, then stacked and fitted into the compression molder (model XH-406B-30-300 from XIHUA Testing Machine Co., Ltd. Guangdong, China). The stacked sheets were compressed with a compression machine at 110 °C and ~10 MPa of pressure for 8 min to obtain an EVA solid pre-form. The EVA pre-form was then compressed at 175 °C with the same pressure for 35 min to complete the foaming process. For comparison, EVA foam incorporated with different contents of talc based on commercial formulations was also prepared with a similar production method. All produced foams were stored in a laboratory environment for at least 40 h before characterization and testing.

### 2.4. Characterization of Foams

#### 2.4.1. Physical Properties

The density of the unfoamed and foamed EVA samples were measured using a densimeter with 0.001 g accuracy, model GT-KD04-203M, from GESTER Instruments, Fujian, China. The foaming ratio was calculated as the following equation:(1)$$\mathrm{Foaming}\;\mathrm{ratio}=\frac{\rho_{u}-\rho_{f}}{\rho_{u}}$$
where *ρ_u_* and *ρ_f_* are density before foaming and density after foaming, respectively.

The expansion ratio was calculated using the following equation:(2)$$\mathrm{Expansion}\;\mathrm{ratio}(\%)=\frac{L_{f}}{L_{m}}\times 100$$
where *L_f_* and *L_m_* are the lengths of the sample after foaming and the length of the metal mold, respectively.

The compression set was carried out using a compression deformation tester, model GT-KB22, from GESTER Instruments, Fujian, China. The foamed samples were subjected to ~40% compression from their original thickness with a compression set device under constant deformation and placed into a carbolite laboratory oven at 45 °C for 6 h. The compression set value (%) was calculated according to the equation below:(3)$$\mathrm{Compression}\;\mathrm{set}(\%)=\frac{t_{0}-t_{f}}{t_{0}-t_{n}}\times 100$$
where *t*_0_, *t_f,_* and *t_n_* are the original thickness of the foam, the final thickness of the foam, and the thickness of the spacer bar used, respectively.

#### 2.4.2. Mechanical Properties

The surface hardness measurements were investigated according to the ASTM D2240 standard with a Shore-C hardness device from GESTER instruments, model GT-KD09-LX-C. The reported hardness data are an average of five measurements from different areas of the foam.

The compression properties of the foams were examined with an LTM5 electro-dynamic testing machine, from Zwick Roell. The test was conducted with a crosshead speed of 1 mm/min on 8 × 8 cm of the samples. All samples were subjected to static compression force until the moving plate reached 50% of the foam’s thickness.

#### 2.4.3. DIN Abrasion Test

The abrasion wear resistance of the CNT/EVA foams was measured according to the DIN-53516 abrasion test standards. The test was conducted with a rotary drum DIN abrasion tester from GESTER instruments, model GT-KB03, China. Cylindrical-shaped samples with a diameter of 16 mm were cut from each foam and mounted in the test holder with 2 mm exposure. The samples were tested against an abrasive paper counterface (60 grits) with a 5 N load. The weight loss of the samples was obtained from the difference between the initial weight (before the test) and the final weight (after the test) of the samples. Wear volume loss (in mm^3^) was calculated by dividing the obtained weight loss by the density of each sample.

#### 2.4.4. Surface Resilience

The surface resilience of the foams was tested according to the DIN 53512 standard using a resilience testing machine from GESTER instrument, model GT-KB18, Fujian, China. A 0.5 Joule impact pendulum was released from a 90⁰ angle and struck into a mounted square foam (8 cm × 8 cm). The maximum rebound heights of each sample were measured based on the reflected gauge meter after impact and the value was recorded after 3 times of warm-up impact strikes from the pendulum. At least 5 measurements were recorded for each sample and the corresponding mean and standard deviation were reported.

### 2.5. Energy-Controlled Dynamic Impulse Testing

#### 2.5.1. Dynamic Impact Test

The dynamic impact and rebound performance of the developed EVA foams were tested using an LTM5 electro-dynamic testing machine with a drive based on linear motor technology from Zwick Roell. This machine is specifically designed to evaluate materials fatigue and endurance performance under both dynamic and static conditions. The test was carried out according to the ASTM F1614 standard, impulse and fatigue of athletic footwear using an energy control piston with a compression head of ~50 mm (Figure 1a). The dynamic loading test regime was designed and adjusted according to an average of athletes’ weight and distance travel under normal training conditions that impacting on the footwear. The piston with energy-controlled of various applied loads of 1.0, 1.25, and 1.5 kN were exerted on the foams (80 mm × 80 mm × 17 mm) at 105 cycles of dynamic impact. These impact forces resulted in approximately 100 kg of body weight compressing towards the tested samples. Figure 1b illustrated the resulting impacted force-cycles test pattern of an EVA foam. The cycle was designed at 3 phases, i.e., initialization stage, (first 5 cycles with the piston frequency of 3 Hz), warming up stage (next 95 cycles with the piston frequency of 2 Hz), and measuring stage (last 5 cycles with the piston frequency of 3 Hz). The force–displacement (*F–d*) data were extracted from the highest points of the measuring phase and the resulting force–displacement F–d curves are employed in the determination of energy returned and energy absorbed by the foams. The energy absorbed by the foams, or the energy lost (hysteresis) is represented through the area between the loading and unloading curves of the force–displacement curve, while the energy returned by the foams is indicated from the area under the unloading curve of the force-displacement curve (Figure 1c). The dynamic response of the developed foams during the dynamic impact test were extracted and analyzed.

#### 2.5.2. Dynamic Fatigue Test

Imitation of a real person’s long-duration running pattern on the foam was conducted using the same LTM5 electro-dynamic testing machine from Zwick Roell. Based on our previous test results, the dynamic fatigue test of the EVA foams that tested on a 20 km distance reflected a similar energy return pattern to the 100–200 km cycles fatigue test. Therefore, the fatigue test of the sequence regime was designed based on 20 km cycles on the foam which is enough to observe and differentiate the performance of the material among the tested samples. The dynamic fatigue test was designed in two stages, i.e., warming-up phase and measuring of the running stage. After the warming-up phase, the foams were subjected to 10,000 cycles with 2 Hz of 1.5 kN of control impact energy, and the test data were extracted every 10 cycles. Repeated foot strikes with approximately ~1.5 Hz are enough to cause fatigue damage to the foam [24]. For performance comparison, neat EVA, selected CNT/EVA foams, and commercial talc-based EVA foams were evaluated.

### 2.6. Scanning Electron Microscopy

The EVA foams’ morphology and cell sizes were observed using a scanning electron microscopy (SEM) on the cryogenic fracture surfaces. The cell density Nc, (cells/cm^3^) was calculated based on the following equation [25,26]:(4)$$\mathrm{Nc}={\left(\frac{n}{A}\right)}^{\frac{2}{3}}$$
where *n* is the cell number in the SEM micrograph, and *A* is the area of the micrograph from the respective sample.

## 3. Results and Discussion

### 3.1. Characterization of the Plastics-Derived CNTs

Properties and characterization of the plastic-derived CNTs are summarized in Table 1 and Figure 2, respectively. The CNT had outer diameters of 10–40 nm for acid-purified CNT (A-CNT) and 10–30 nm for air-treated oxygenated CNT (O-CNT). The length of CNTs reached several micrometers as suggested by the FESEM image in Figure 2a. Both samples were characterized by type II isotherms with negligible hysteresis loops (Figure 2b). However, O-CNT had larger BET specific surface area and total pore volume compared to A-CNTs. The distinctive D- and G-bands at ~1350 and 1560 cm^−1^ specific to carbon samples can be observed in the Raman spectra (Figure 2c). D-band corresponds to defective, while G-band to graphitic regions in carbon crystals. The degree of graphitization can be quantified by the ratio of areas of D- and G-bands. According to Table 1, the ID/IG ratio for O-CNTs was lower, suggesting the lower density of defects, probably, due to the use of high temperature during the purification stage. On the contrary, the oxidation degree was higher as suggested by the greater oxygen content on the surface of O-CNT. Wide scan XPS spectra show that only two elements, C and O, were detected by the instrument on the surface of CNT samples (Figure 2d). The core C1s XPS spectra are shown in Figure 2e and disclose a variety of carbon bonds, corresponding to metal carbide (282.5 eV), C=C in graphitic carbon (284.9 eV), C-O in phenols, pyran, ether and alcohols (286.9 eV) and π-π* bonds in carbon [27,28]. The lower ash content of O-CNT compared to A-CNT (0.4 wt% against 3.0 wt%) suggests better efficiency of high-temperature purification via chlorination compared to acid leaching of the catalyst.

### 3.2. Effect of Different Kinds of CNTs on EVA Foam’s Properties

The study first compared the effect of two different kinds of CNTs on the properties of EVA foams. The physical and mechanical properties of the EVA incorporated with O-CNT and A-CNT at the filler content of 0.25 and 0.50 wt% were shown in Table 2. The density of the EVA foams filled with A-CNT was found to be slightly lower than the O-CNT for both filler content. In terms of the physical and mechanical properties, the O-CNT/EVA foams exhibited greater surface hardness, resilience, and compression strength compared to A-CNT/EVA foams. This could be due to the better compatibility of O-CNT with the EVA which provides additional mechanical strength and stiffness to the foams. It was reported that the melt blending of EVA with various types of modified montmorillonite nano-fillers affects the mechanical properties and cellular structure of the EVA foams [29].

The effect of different kinds of CNTs on the dynamic impact properties of EVA foams was shown in Figure 3. It was found that the O-CNT/EVA foams (CNT-025-O and CNT-050-O) displayed a larger peak and area under curves as compared to the A-CNT/EVA foams (CNT-025-A and CNT-050-A). The average energy return of the CNT/EVA foams at various control impact loads of 1.0, 1.25, and 1.5 kN were presented in Table 3. The EVA incorporated with O-CNT shows higher energy return (greater foam rebound capability) than the A-CNT for all three control impact forces. The O-CNT/EVA shows approximately 2–4% greater in terms of energy return response of the foam than the A-CNT/EVA foams at all applied loads of 1.0, 1.25, and 1.50 kN. In addition, the energy absorbed by the A-CNT/EVA foams is larger than the O-CNT/EVA foams (Figure 3b), which implies greater energy loss and energy dissipation of the foam when subjected to dynamic impact forces.

As presented in the BET and XPS analysis results in Table 1, the O-CNT exhibits higher specific surface area and oxygen functional groups. The improved physical, mechanical, and energy return properties of the foam by the O-CNT could be due to several reasons. The higher available specific surface area and surface functionalization increased interacting surfaces between the O-CNT and EVA, which prevent large energy loss. In a reported study on functionalized CNT/PU foams, substantial improvement in terms of thermal, flexural, acoustic properties, and compressive stability was observed by the oxidized CNT over as-grown CNT [8]. Therefore, the presence of oxygen functional groups of CNT could promote a greater extent of interactions with EVA. As well as the surface interactions, the A-CNT have a bulk density compared to O-CNT, as shown in Table 1. This indicates it occupies a smaller volume, which could be a sign of agglomeration and, hence, lowers the dispersibility and performance when incorporated in EVA. The bulk density, degree of graphitization, specific surface area, and surface functionalization of CNT could potentially affect and determine the EVA foam properties.

Based on the superior performance of the O-CNT/EVA foams as compared to the A-CNT/EVA foams, only the oxygenated O-CNT was used for further exploration of the EVA foam performance at various content of 0.05 to 1.0 wt%. All the CNT specified started from the subsequent section are all from the oxygen-functionalized CNT.

### 3.3. Physical and Mechanical Properties of CNT/EVA Foams

The physical properties of CNT/EVA foams were presented in Table 4. The neat EVA foam exhibits a soft surface hardness and low density with an average value of 0.13 g/cm^3^. The incorporation of CNT increased the density and decreased the foaming ratios of the EVA foams. This suggests the presence of CNT restricted the foaming and expansion of the foams. The compression set of the EVA was found to improve after the incorporation of CNT. The lower percentage of the compression set, the better the elastic recovery of the foam and better retention of its elastic properties after long hours of static compression. EVA foam is having poor compression set properties with more than 50% [14,30]. The compression set of EVA foam was found to improve from ~59% to ~55% after the addition of CNT. It was reported that the compression set of EVA foam was improved by 30% with the incorporation of MWCNT at 5 phr [14]. The filler–matrix interactions in the composite foams play a vital role in the elastic recovery properties. For instance, the compression set of the EVA/ethylene-1-butene copolymer blend foams was found to reduce after the incorporation of organoclay. It can be associated with the poor interactions between the hydrophilic clay and hydrophobic EVA, which results in the polymer chain slipping along the interacting surfaces and causing energy dissipation [31]. The chain slipping could be prevented through a stronger chemical bonding between filler and matrix in the nanocomposites, hence, improving the compression set. It was reported that the presence of the CNT lowers the mobility of the polymer chains [16]. In our results, the improved compression set implies the strong interphase connection between the oxygenated functionalized CNT and EVA nanocomposites.

The surface hardness (shore C) of the EVA was increased from approximately 28 to more than 30 after the addition of CNT. On the other hand, EVA foam exhibited a better surface resilience as compared to the CNT/EVA nanocomposite foams. The reduction in the surface rebound resilience could be possibly due to the increased energy absorption of the CNT/EVA foams compared to the neat EVA foam. It was also reported that the rebound resilience of the EVA foam was reduced with the increasing content of nano-organically modified montmorillonite [32]. Studies have shown that surface resilience usually compromises with the increase of surface hardness which is confirmed by the results obtained [33].

Figure 4a–d showed the expansion ratios (ER), abrasion resistance, and compression properties of the CNT/EVA foams. As shown in Figure 4a, the CNT/EVA nanocomposite foams showed lower ER than the neat EVA foam. The ER of the CNT/EVA foams decreased with increasing content of CNT up to 0.75 wt% and increased again at 1.0 wt% of CNT loading. A similar trend was observed with the density and foaming ratios of the CNT/EVA foams. This confirms the enhanced interactions between the CNT and EVA and improved? viscosity which affects the expansion of the cells. The abrasion resistance of the EVA foam was improved after the incorporation of CNT as can be seen in the reduction of the wear volume loss of the CNT/EVA foams compared to the neat EVA foam (Figure 4b). The compression stress of the EVA foam was also found to enhance with the presence of CNT (Figure 4c,d). The 0.5 wt% CNT/EVA foam exhibited the highest compression strength. The improvement in the abrasion resistance and compression strength could be due to the decent reinforcing effects of the oxygen functionalized CNT in the EVA. CNT with different aspect ratios has also been shown to improve the compressive properties of PMMA nanocomposite foams [11].

### 3.4. Dynamic Impact Response and Energy Return

The CNT/EVA foams were subjected to a dynamic impact load test and the extracted dynamic impact properties and energy return performance were shown in Figure 5a–f. From the normalized force and displacement curves in Figure 5a, all the CNT/EVA foams exhibit a typical loading and unloading curve with narrow hysteresis loss of energy. Despite the good foam properties with low energy loss, EVA foam without reinforcement is lacking mechanical properties, as can be seen in the previous mechanical properties section. The dynamic impact test for the EVA foam cannot be performed above 1.5 kN of applied load due to its softness and low compression strength. On the dynamic impact response properties, the dynamic stiffness of EVA increases with the increasing content of CNT (Figure 5b). The EVA foam shows greater energy stored, and energy loss when compared to CNT/EVA foams (Figure 5c,d). In addition, the EVA foams required a longer recovering time after impact throughout the test cycles, as can be seen in the linear increase of the loss factor curve of EVA from the beginning of the test cycle (Figure 5e). Meanwhile, the CNT/EVA foams exhibit short recovering time upon impact with a relatively constant loss factor curve after the pressure applied. This indicates the softness of EVA that prolongs the impact absorption cycles upon impact. The reduction of stored energy, loss energy, and loss factor of the CNT/EVA foams was observed as the content of the CNT is increasing with the highest at 1.0 wt%. The presence of CNT in polymer foams leads to stiffer foams and hence, lower energy dissipation under cyclic compression in comparison to neat polymer foams [8]. Figure 5f shows the calculated value of the area under the unloading curve of the force-displacement graphs which represents the energy return of the foam. It was found that the energy returns of the EVA foam increase with the increasing content of the functionalized CNT up to 0.5 wt%. This shows that the presence of nano-scale CNT played a significant role in improving the energy rebound of the EVA foam. The energy return of the foam started to decline when the CNT content was above 0.5 wt%. As presented in the previous section, the EVA foam with 0.5 wt% CNT also revealed the highest compression set and compression strength. This indicates that 0.5 wt% loading of CNT provides optimum performance of the EVA foam in terms of filler–matrix interactions, filler dispersion, and cell structures. Further addition of the CNT higher than 0.5 wt% could disrupt the intrinsic balance properties of the CNT/EVA nanocomposites foam and, hence, affect the overall performance of the composites.

### 3.5. Cell Morphology of CNT/EVA Foams

The SEM micrographs of the CNT/EVA foams at different content of CNT were presented in Figure 6 and their respective calculated cell density values are shown in Figure 7, respectively. It was observed that the cells’ shapes and cells’ structures of the EVA are not significantly changed with the incorporation of CNT. However, the cell sizes of the EVA were reduced with the increasing content of CNT. The cell sizes of the foam were reduced significantly from 0.75 wt% of CNT to 1 wt% of CNT content in EVA (Figure 6e,f). The reduction of cell size correlated well with the calculated cell density. The cell density is increasing as the amount of CNT is increased in the foam (Figure 7). At 1 wt% of CNT, a noticeable increase in cell density, and a decrease in cell sizes were observed as compared to the neat EVA and the other CNT/EVA foams. This could be due to the domination of heterogeneous nucleation to promote blowing and cell nucleation as the amount of CNT is increasing to 1 wt%. Smaller cell size and higher cell density of the 1 phr MWCNT/EVA foam as compared to EVA foam were also reported by Yu and Kim [30]. High aspect ratio nano-fillers such as organically modified montmorillonite was also reported to be able to improve the uniformities of the cell structure and reduction of cell size of the EVA due to the strong heterogeneous nucleation effect imparted by the nano-fillers [29,32].

The increase in the cell density and decrease of the cell sizes of 1.0 wt% CNT/EVA foam was corroborated with the reduction in the density of the CNT/EVA foams presented in the previous section on the physical properties of the foams, where the density was reduced from 0.75 wt% of CNT to 1.0 wt% of CNT content. The cells size of the PMMA nanocomposite foams was also found to reduce after the incorporation of CNT [11]. However, there is no substantial positive increment in terms of mechanical properties and energy return of the EVA foams when the foam’s cell sizes were reduced at 1 wt% of CNT-filled EVA foam as shown in the previous section of hardness, compression strength, and energy return results.

### 3.6. Comparison with Conventional talc/EVA Foam

To further investigate the potential of the CNT/EVA foam for sport shoe midsole application, the energy returned acquired from the dynamic impact response of the CNT/EVA foams were compared with the existing established talc-based EVA foams formulations that are commonly used for footwear midsole production. Figure 8a,b shows the energy return at various applied loads of different content of talc and CNT-reinforced EVA foams, respectively. The energy return trend of the foams was found to be similar for the three applied loads, i.e., 1.0, 1.25, and 1.50 kN. Higher energy return value was observed with higher piston force applied from 1.0 to 1.50 kN. This is due to the higher rebound force being reflected by the tested foams as a higher impact force is applied. From the extracted energy return results of talc/EVA, it was noticed that the energy return of EVA was reduced by about ~10% with the incorporation of 5 wt% talcs (Figure 8a). The energy return of the EVA increased linearly from 10 to 15 wt% of talc in EVA, and then reduced drastically at 20 wt%. Despite the increasing trend of energy return observed from 5 to 15 wt% of talc loading, the energy return of the talc/EVA foams are all lower than the neat EVA foam.

On the other hand, the energy return of EVA foam was found to increase after the incorporation of CNT from 0.05 to 0.75 wt% (Figure 8b). Highest energy return was observed at 0.25 and 0.5 wt% of CNT content in EVA. The energy return of the CNT/EVA shows decreasing trend at 1.0 wt% of CNT loading. Significant improvement on the cyclic compression recovery of the thermoplastic foams after the addition of the optimum amount of CNT has been also reported by other researchers [34,35]. From the dynamic impulse test, it was found that all the CNT/EVA foams showed better energy return as compared to talc/EVA foams. This indicates that the replacement of talc mineral fillers with a small amount of nano-filler, e.g., CNT as reinforcing agent and nucleating agent could enhance the mechanical properties of EVA foam without compromising its energy return performance.

Table 5 compared the overall properties profile between the optimum energy return of CNT/EVA and talc/EVA foams. At the 1.5 kN of impact force, 0.5 wt% CNT/EVA and 15 wt% talc/EVA foams exhibited the highest energy returned value. Compared to the talc/EVA foam, the CNT/EVA foam surpassed most of the talc/EVA foam’s properties which are desired for footwear midsole application. The CNT/EVA exhibited a lighter weight, higher surface hardness, compression strength, specific compression strength, compression set, and energy return as compared to talc/EVA. In addition, the energy absorption or energy loss of CNT/EVA is lower than that of talc/EVA foam. This could be due to the nano-reinforcing effects of the nano-sized CNT. Nano-fillers can improve both the stiffness and energy absorption of polymers, especially nano-fillers that are having large aspect ratios. The interfacial shear strength between the nano-filler and matrix is higher than in conventional composites due to the formation of cross-links or supramolecular bonding which cover or shield the nano-fillers and form thicker interphases than in conventional composites [36].

Compressive stress is one of the important attributes of foam materials for resistance to compression force. Sufficient compressive stress is required for EVA foam used in the shoe’s midsole application in order to sustain body loads without collapsing the cells. Figure 9 presents the compression stress of the EVA, CNT/EVA, and talc/EVA foams at the ranged of the filler contents used in this work as a function of the density of the foams. In general, it was found that the compressive strength of the EVA foams is dependent on the density. The compressive stress of the EVA foam increases with the increase in density. The ideal performance parameters for a good footwear are high compressive stress and low density for durable and light-weight shoe. However, these two parameters are usually compromised with each other. Low-density foam is usually exhibits lower compression strength as compared to high-density foam. The closed-cell polymer foams’ energy absorption and peak compression stress are strongly dependent on its density [37]. With the addition of a small amount of CNT nano-additives from the range of 0.05 to 1.0 wt%, the nucleation and blowing effect of the EVA were comparable to the addition of 5 to 20 wt% of talc, where both of their densities were in parallel range. The EVA foams remain lightweight after the incorporation of CNT and talc, where the density is not exceeding 0.2 g/cm^3^. It is interesting to note that the CNT/EVA foams are imparting the next level of compressive strength as compared to talc/EVA foams at a comparable density. Approximately 20% enhancement in compressive strength was observed for the CNT/EVA foams as compared to the talc/EVA foams.

### 3.7. Dynamic Fatigue Test of CNT/EVA Foam

The dynamic fatigue test envisages the long-term durability of shoe foam. Through the thickness loss of the foams from the control cyclical impact force during the fatigue test, the long-term durability performance and behavior of the foam can be predicted. The fatigue of the foam could cause running injuries due to the reduction of heel strike cushioning [24]. Figure 10 shows the 10,000 cycles of dynamic fatigue test curves of EVA, talc/CNT (15 wt%), and CNT/EVA (0.5 wt%) foams. The lower measured displacement indicates a larger thickness loss of the foam. It was found that CNT/EVA foam exhibits greater resistance to fatigue with lower thickness loss as compared to EVA and talc/EVA foams. The trend follows by the talc/EVA foam and lastly neat EVA foam. The retention of the thickness loss by the three tested samples shows a larger difference between each other with the increasing number of cycles (running distance) of the fatigue test. This indicates that the presence of a small amount of CNT could preserve the foam characteristic and make it more durable which in turn improves longevity and the quality of the foam. The dynamic fatigue test correlates well with the mechanical properties and energy return characterization whereby the performance of CNT/EVA is superior to the EVA and talc/EVA foams. The CNT/EVA foams demonstrated many favorable property profiles against the conventional mineral-filled EVA foams, which justifies future research of nano-additives for higher energy rebound EVA foam development.

## 4. Conclusions

This work demonstrated EVA nanocomposite foams with balanced properties of mechanical and energy return through the incorporation of a multi-walled CNT-derived from plastics. The prepared two kinds of CNT derived from the upcycling of commodity plastics resulted in varying physical and mechanical properties of the CNT/EVA foams. EVA foam incorporated with oxygenated CNT displays greater surface hardness, surface resilience, compressive strength, and energy return in comparison to acid-purified CNT. The incorporation of oxygenated CNT improved the surface hardness, abrasion resistance, compression set, and compressive strength of the EVA foam. The dynamic stiffness, energy storage, and energy return of EVA were enhanced after the incorporation of oxygenated CNT. The foams’ compression strength, compression set, and energy return show the highest improvement at 0.5 wt% of CNT content. The presence of CNT in EVA resulted in higher cell density and smaller cell size. The CNT/EVA foams provide a lighter-weight solution of foams with approximately ~20% increment in compressive strength when compared to talc/EVA foams. From the acquired dynamic fatigue results, CNT/EVA is better at retaining the foam’s structures after 10,000 cycles of dynamic impact as compared to the neat EVA and talc/EVA foams.

Overall, CNT appeared as an effective reinforcing agent and nucleating agent for EVA foam. It is possible to replace conventional mineral fillers with a small amount of CNT (<1.0 wt%) to counter the energy rebound loss of EVA foam and at the same time improve its mechanical properties. The developed mechanical-energy return balanced CNT/EVA foams display great potential for midsoles of high-performance running shoes, which could improve the running performance of the athletes. Though the two different purification and processing methods of CNT presented in this work could affect the EVA foams’ properties, other purification strategies and surface functionalized CNT are required to investigate in order to confirm the main factors that contribute to the energy return improvements of the foam.

## Figures and Tables

**Figure 1 polymers-15-00948-f001:**
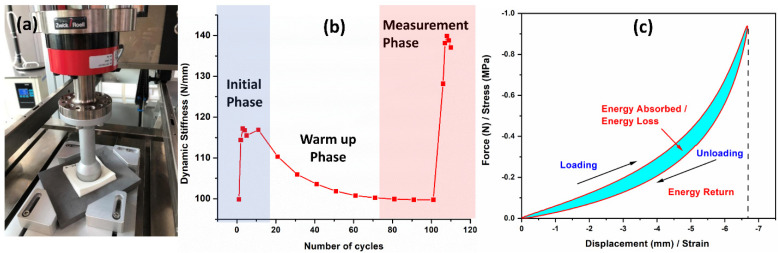
(**a**) The operation of the Zwick Roell LTM5 electro-dynamic testing machine on the foam samples. (**b**) The designed dynamic testing cycles profiles and (**c**) the typical force-displacement curve of the dynamic compression test and the energy representation from the curve.

**Figure 2 polymers-15-00948-f002:**
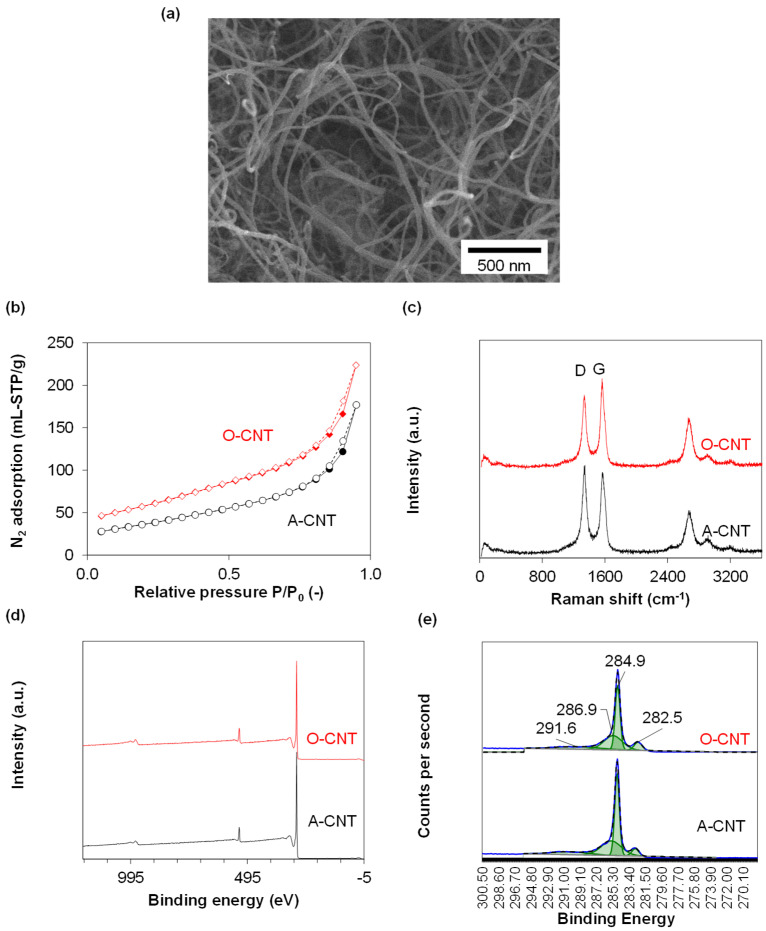
(**a**) Representative FESEM image of the O-CNT sample, (**b**) N2 adsorption-desorption isotherms and (**c**) Raman spectra of the CNTs, (**d**) wide scan XPS spectra and (**e**) core C1s XPS spectra.

**Figure 3 polymers-15-00948-f003:**
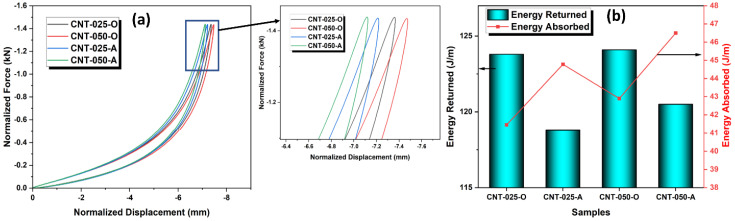
Dynamic cyclic impact results of the two variants of CNTs-reinforced EVA nanocomposite foams at control impact force of 1.5 kN. (**a**) Force-displacement curves and (**b**) energy returned and energy absorbed.

**Figure 4 polymers-15-00948-f004:**
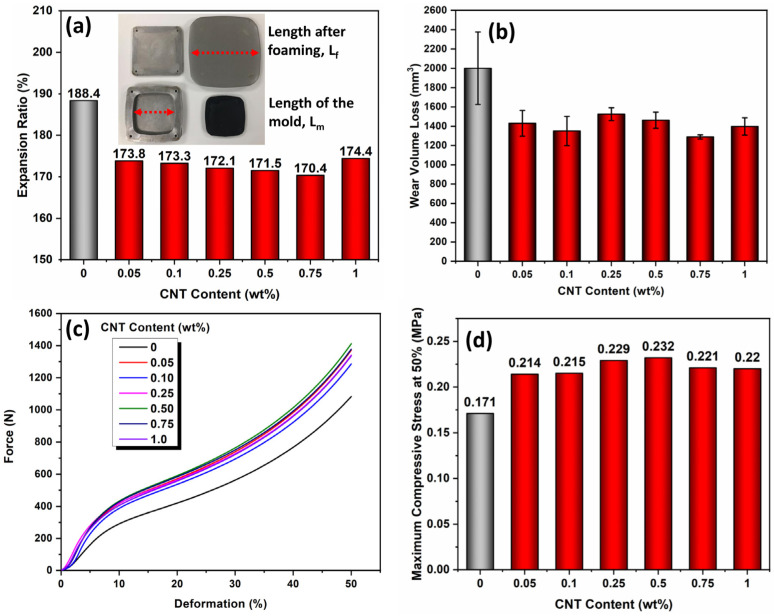
(**a**) Percentage expansion ratio, (**b**) DIN abrasion wear loss, (**c**) static compression curves and (**d**) maximum compression strength of the EVA and CNT/EVA nanocomposite foams.

**Figure 5 polymers-15-00948-f005:**
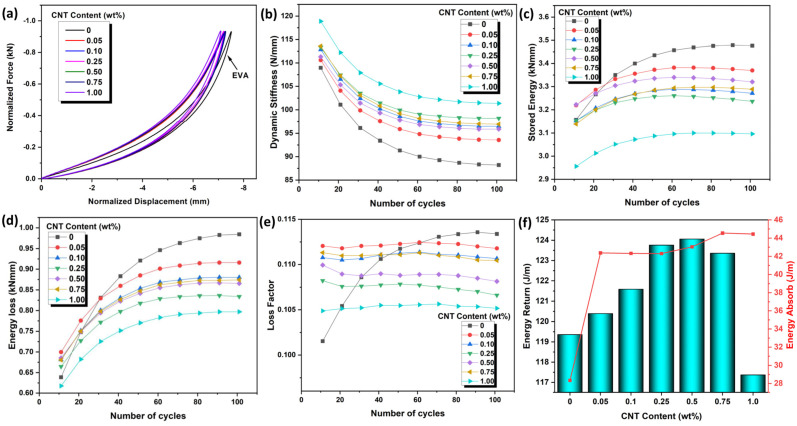
Dynamic impact response of CNT/EVA foams at the control impact force of 1.5 kN, (**a**) normalized force-displacement curves, (**b**) dynamic stiffness, (**c**) stored energy, (**d**) energy loss, (**e**) loss factor and (**f**) energy returned and energy absorption.

**Figure 6 polymers-15-00948-f006:**
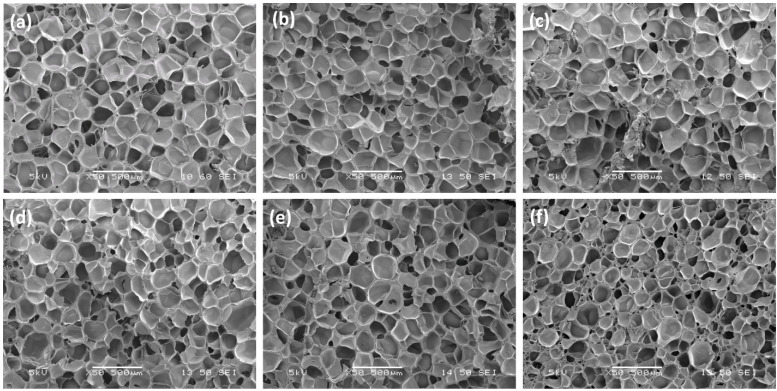
SEM micrographs of the fractured surfaces of (**a**) EVA foam, (**b**) 0.1 wt%, (**c**) 0.25 wt%, (**d**) 0.50 wt%, (**e**) 0.75 wt% and (**f**) 1.0 wt% of CNT/EVA foams.

**Figure 7 polymers-15-00948-f007:**
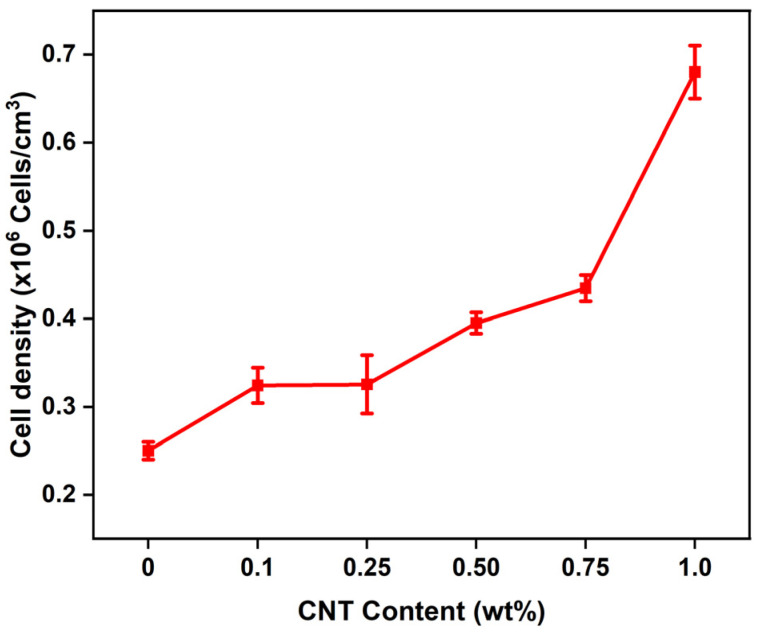
Cell density of the CNT/EVA foams as a function of CNT content.

**Figure 8 polymers-15-00948-f008:**
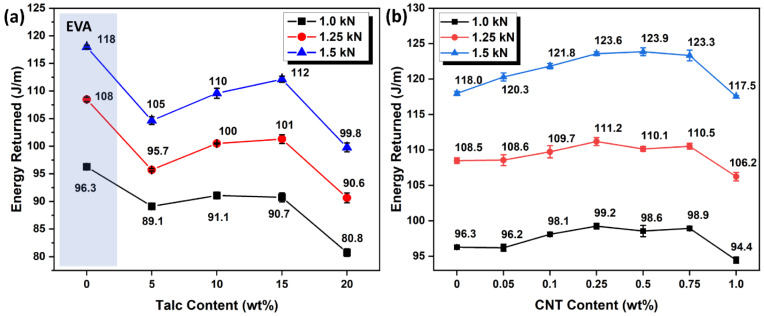
Energy returned of (**a**) talc/EVA foams and (**b**) CNT/EVA foams at different content and applied loads.

**Figure 9 polymers-15-00948-f009:**
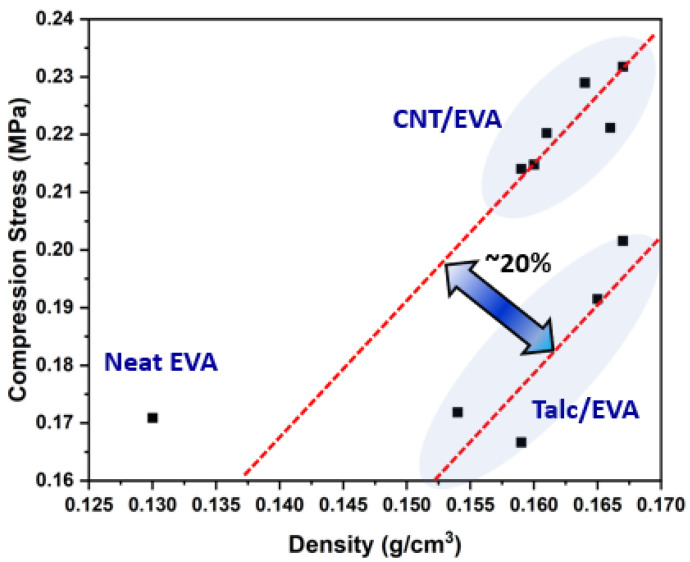
The compression stress versus density chart of the neat EVA, CNT/EVA, and talc/EVA foams.

**Figure 10 polymers-15-00948-f010:**
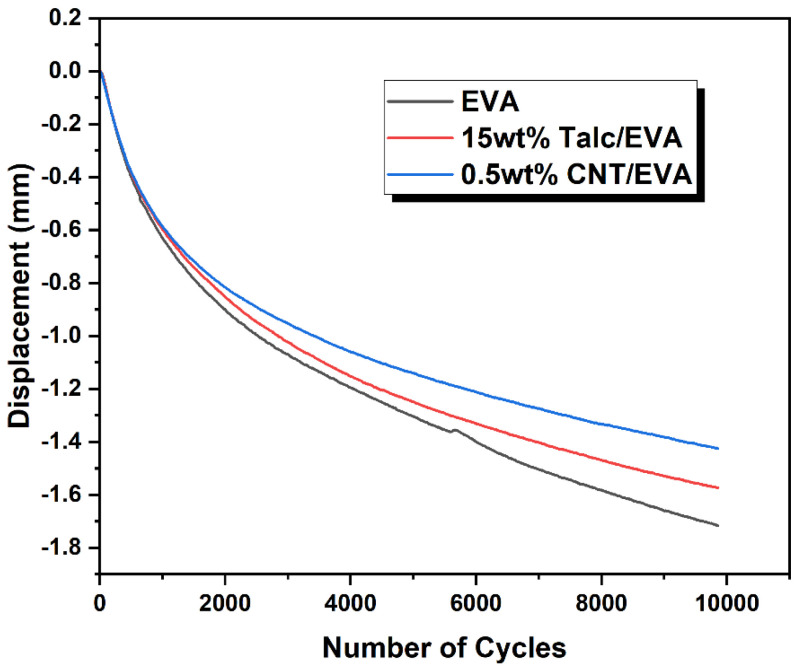
Displacement versus the number of cycles curves from the dynamic fatigue test for EVA, CNT/EVA, and talc/EVA foams.

**Table 1 polymers-15-00948-t001:** Characteristics of CNTs.

Properties	A-CNT	O-CNT
Bulk density (g/cm^3^)	0.102	0.067
Outer diameters (nm)	10–40	10–30
BET specific surface area (m^2^/g)	131	205
Total pore volume (cm^3^/g)	0.27	0.35
ID/IG (-) ^a^	1.14 ± 0.11	0.87 ± 0.03
C (wt%) ^b^	93.30 ± 0.30	90.44 ± 0.38
O (wt.%) ^b^	6.7 ± 0.30	9.56 ± 0.38
Ash (wt%)	3.0	0.4

^a^ calculated as average ± standard deviation of three measurements from different sample areas; ^b^ from XPS analysis.

**Table 2 polymers-15-00948-t002:** Physical and mechanical properties of the different kinds of CNT-reinforced EVA foams.

CNT/EVA Foam	Density (g/cm^3^)	Hardness (Shore C)	Resilience (%)	Compression Stress (MPa)	Specific Compression Stress (MPa/g-cm^3^)
CNT-025-O	0.164	35.6 ± 1.0	58.4 ± 0.42	0.229	1.431
CNT-050-O	0.167	35.8 ± 1.4	57.5 ± 0.50	0.232	1.440
CNT-025-A	0.157	32.1 ± 0.7	56.4 ± 0.38	0.209	1.351
CNT-050-A	0.161	32.9 ± 1.2	56.7 ± 0.41	0.208	1.307

**Table 3 polymers-15-00948-t003:** Comparison of average energy return of different kinds of CNT-filled EVA foams under the dynamic applied load of 1.0, 1.25, and 1.50 kN.

CNT/EVA Foam	Average Energy Return at 1.0 kN (J/m)	Average Energy Return at 1.25 kN (J/m)	Average Energy Return at 1.50 kN (J/m)
CNT-025-O	99.2	110.9	123.8
CNT-025-A	95.5	107.5	118.8
Percentage difference (%)	3.87	3.16	4.21
CNT-050-O	98.6	110.1	124.1
CNT-050-A	95.4	107.7	120.5
Percentage difference (%)	3.35	2.23	2.99

**Table 4 polymers-15-00948-t004:** Physical properties of the CNT/EVA foams.

CNT/EVA Foam	Density (g/cm^3^)	Foaming Ratios	Compression Set (%)	Hardness (Shore C)	Resilience (%)
EVA	0.130	86.53	59.01	28.3 ± 1.1	59.0 ± 0.32
0.05 CNT/EVA	0.159	83.23	54.73	33.0 ± 0.9	57.1 ± 0.22
0.10 CNT/EVA	0.160	82.88	54.92	34.1 ± 1.4	57.2 ± 0.27
0.25 CNT/EVA	0.164	83.01	54.33	35.6 ± 1.0	58.4 ± 0.42
0.50 CNT/EVA	0.167	82.99	53.43	35.8 ± 1.4	57.5 ± 0.50
0.75 CNT/EVA	0.166	82.73	54.17	36.9 ± 1.7	57.4 ± 0.42
1.00 CNT/EVA	0.159	83.58	55.78	35.2 ± 1.4	56.5 ± 0.27

**Table 5 polymers-15-00948-t005:** Properties profile comparison of CNT/EVA foams to commercially available talc-based EVA foams.

Property	Unit	CNT/EVA (0.5 wt%)	Talc/EVA (15 wt%)
Density	g/cm^3^	0.161	0.165
Hardness	Shore C	35.8 ± 1.4	33.3 ± 0.6
Compression Strength	MPa	0.232	0.202
Specific Compression Strength	MPa/g/cm^3^	1.440	1.222
Compression Set	%	53.43	57.42
Energy Returned at 1.5 kN	J/m	123.9	112
Energy Absorbed at 1.5 kN	J/m	43.0	46.4

## Data Availability

The data supporting the findings of this manuscript are available from the corresponding authors upon reasonable request.

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
