# Peer review of "Nanocomposite Foams with Balanced Mechanical Properties and Energy Return from EVA and CNT for the Midsole of Sports Footwear Application"

_polymers, 2023, doi:10.3390/polym15040948_

Round 1

Reviewer 1 Report

A good and comprehensive study of the development of EVA-CNT composites for foaming applications and the associated morphology and macroscopic performance, well-presented and with logical conclusions. Only some minor fine-tuning suggestions below before; otherwise publication is supported.

Title: This is the only instance where the abbreviation MWCNT is used for the carbon nanotubes of this work (MWCNT is used herein only for references where this nomenclature is used). I would recommend to change it to CNT for alignment with the rest of the text.

Line 72: elevating --> elevate

Lines 186 & 252: The "-1" and "3" needs to be set as exponent, respectively.

Lines 116, 118, 143: A space is needed between the value and the unit °C.

Figure 4: The legend of panel 4c needs a header "CNT Content (wt%)", i.e. similarly to Figure 5.

Figure 5: In the legends of the CNT Content the unit wt% needs to be added.

Figure 7: The cell density, as calculated by equation 4 and the SEM images of Figure 6 quite definitely contains some error. Probably some error bars would be needed to indicate this?

Author Response

Response to Reviewer’s Comments

Reviewer 1

A good and comprehensive study of the development of EVA-CNT composites for foaming applications and the associated morphology and macroscopic performance, well-presented and with logical conclusions. Only some minor fine-tuning suggestions below before; otherwise publication is supported.

 The authors would like to thank the reviewer for the favorable comment.

Title: This is the only instance where the abbreviation MWCNT is used for the carbon nanotubes of this work (MWCNT is used herein only for references where this nomenclature is used). I would recommend to change it to CNT for alignment with the rest of the text.

Thanks for the constructive comments. As suggested, we have revised and changed the MWCNT abbreviation in the title to CNT. Please refer to our revised manuscript for details. 

Line 72: elevating --> elevate

Thanks for the corrections. It has been revised accordingly. Please refer to our revised manuscript for details. 

Lines 186 & 252: The "-1" and "3" needs to be set as exponent, respectively.

Thanks for the corrections. It has been revised accordingly. Please refer to our revised manuscript for details. 

Lines 116, 118, 143: A space is needed between the value and the unit °C.

Thanks for the corrections. It has been revised accordingly. Please refer to our revised manuscript for details. 

Figure 4: The legend of panel 4c needs a header "CNT Content (wt%)", i.e. similarly to Figure 5.

Thanks for the corrections and suggestions. As suggested, the legend of Figure 4 and Figure 5 have been revised accordingly. Please refer to the revised Figure 4 and Figure 5 in our revised manuscript for details. 

Figure 5: In the legends of the CNT Content the unit wt% needs to be added.

Thanks for the corrections and suggestions. As suggested, the legend of Figure 5 has been revised and added wt%. Please refer to the revised Figure 5 in our revised manuscript for details. 

Figure 7: The cell density, as calculated by equation 4 and the SEM images of Figure 6 quite definitely contains some error. Probably some error bars would be needed to indicate this?

Thanks for the corrections and suggestions. We have conducted additional measurements on the foams’ cell density counts and we have included the error bars /standard deviation in Figure 7. Please refer to the revised Figure 7 in our revised manuscript for details. 

Reviewer 2 Report

Manuscript describes an interesting comparison of properties of EVA foam when traditional talc was substituted by two types of CNT, which act as nucleation agents. This substitution allowed to reach better mechanical properties such as energy return, compression stress, which are required in sport footwear. EVA foam with CNT has interesting combination of improved stiffness and ameliorated rebound energy. All samples were rigorously characterized with large number of tests. Dynamic mechanical properties including fatigue test were measure which is generally rear in material characterization.  In addition, authors clearly stated that different types of CNT may significantly influence properties of composite material. Article is logical, accurate and well written.

A few comments and questions to authors are below.

1.Abstact. Line 24. (and further in the text) Can we say that CNT made of used polymers is upcycling? Formally it is destruction of polymers. Authors should check whether manufacturing of CNT from polymers is fitting to the definition of upcycling. It is not negative comment, it is an open question, whether we can use the word upcycling to destructive transformation of recycled materials.

2.Page 2. Line 81. You provided reference 13, saying that properties of composite EVA plus CNT were studied by Park and Kim. In this case, it would be good to say about novelty in your research devoted to the same subject.

3.Page 3. Line 116. Could authors give a little bit more information about chlorination method used for functionalization of CNT. How was it done? Maybe a reference would be enough.

4.Page 3. Line 137. You indicate about mixing until the homogeneous distribution of components. How did you control it? And what is the approximate mixing time?

5.Page 7. Line 275. How you come out to this level of CNT concentration: 0.25 and 0.5%? Is it literature data or did preliminary testing with different CNT concentration? Why not 1 or 2%?

6.Page 12. Lines 434-435. The authors write about 1% wt of CNT. However, sometimes phr is used. For example, in the section “EVA foam preparation”. Maybe difference between % and phr is small, but it would be good to stick to one expression of the amount. If we look accurately % wt should not correspond to phr.

7.You measured compression set and compressive strength of samples. Is it necessary do both measurements, because both of them characterize the same property?

Author Response

Response to Reviewer’s Comments

Reviewer 2

Manuscript describes an interesting comparison of properties of EVA foam when traditional talc was substituted by two types of CNT, which act as nucleation agents. This substitution allowed to reach better mechanical properties such as energy return, compression stress, which are required in sport footwear. EVA foam with CNT has interesting combination of improved stiffness and ameliorated rebound energy. All samples were rigorously characterized with large number of tests. Dynamic mechanical properties including fatigue test were measure which is generally rear in material characterization.  In addition, authors clearly stated that different types of CNT may significantly influence properties of composite material. Article is logical, accurate and well written.

The authors would like to thank the reviewer for the favorable comment.

A few comments and questions to authors are below.

1.Abstact. Line 24. (and further in the text) Can we say that CNT made of used polymers is upcycling? Formally it is destruction of polymers. Authors should check whether manufacturing of CNT from polymers is fitting to the definition of upcycling. It is not negative comment, it is an open question, whether we can use the word upcycling to destructive transformation of recycled materials.

Thanks for the comments. According to the Merriam-Webster dictionary upcycling is "to recycle (something) in such a way that the resulting product is of a higher value than the original item; to create an object of greater value from (a discarded object of lesser value)". Considering that synthesized CNT product has a higher value than polymers used, we have reasons to believe that the term "upcycling" was used correctly.

Furthermore, the analysis of scientific literature suggests that the term "upcycling" is commonly used in the similar context:

doi.org/10.1002/app.39931

doi.org/10.1038/s41586-021-04350-0

2.Page 2. Line 81. You provided reference 13, saying that properties of composite EVA plus CNT were studied by Park and Kim. In this case, it would be good to say about novelty in your research devoted to the same subject.

Thanks for the comments. The novelty of this work has been described in the next paragraph after line 81. “Though, most of the polymer filled with CNT composite foam studies thus far are not focusing on the dynamic impulse rebound (energy return) and the dynamic response of the foam, particularly for EVA foam used for sportswear midsole application.” “In this work, we investigate the physical, mechanical, dynamic impact properties and dynamic fatigue performance of EVA filled with different kinds of CNTs derived from upcycling of commodity plastics.”

3.Page 3. Line 116. Could authors give a little bit more information about chlorination method used for functionalization of CNT. How was it done? Maybe a reference would be enough.

Thanks for the comment and suggestion. As suggested, the reference was provided in the revised version: "To functionalize CNTs two methods were used. Oxygenated CNTs (O-CNT) were prepared using chlorination above 1000 °C with a modified method from [doi.org/10.1039/C3CP50348H] and subsequent treatment with air described in [doi.org/10.1007/s10853-016-9864-0]." Please refer to page 3 of our revised manuscript for details.

4.Page 3. Line 137. You indicate about mixing until the homogeneous distribution of components. How did you control it? And what is the approximate mixing time?

Thanks for the comments. The homogeneous distribution of components within the EVA is indicated by the uniform texture and color of the rolled masticated EVA sheet after the high-speed mixing in the two-rolled mill. The details of mixing conditions and the appropriate mixing time were added in the revised manuscript accordingly. Please refer to page 3 of our revised manuscript for details.

5.Page 7. Line 275. How you come out to this level of CNT concentration: 0.25 and 0.5%? Is it literature data or did preliminary testing with different CNT concentration? Why not 1 or 2%?

Thanks for the comments. CNT is a nanomaterial in the range of less than 100 nm. Because of their large effective surface area, nanomaterials are usually used in very small amounts in polymer composite development. Most of the published literature has shown that a small amount of nano-size fillers was able to improve the properties of the polymer nanocomposites. The aim of using a small amount of concentration of 0.05 to 1.0 wt% is to avoid large agglomeration of nanomaterials (CNT in this case) in polymer nanocomposite foam. In addition, nanomaterials are usually costly due to their complex synthesis and purification process, the use of a small amount of nanofillers is to reduce the overall cost of the final developed product (mid soles for the high-performance shoe in our work).

6.Page 12. Lines 434-435. The authors write about 1% wt of CNT. However, sometimes phr is used. For example, in the section “EVA foam preparation”. Maybe difference between % and phr is small, but it would be good to stick to one expression of the amount. If we look accurately % wt should not correspond to phr.

Thanks for the comments and corrections. We have checked and revised accordingly with all the units consistent in wt% throughout the entire manuscript. The phr unit in line 332 and line 419 are from the work cited in references 13 and 27 respectively. Please refer to our revised manuscript for details.

7.You measured compression set and compressive strength of samples. Is it necessary do both measurements, because both of them characterize the same property?

Thanks for the comments. We agreed with the reviewer that there is a small extent of similarity between compression set and compressive strength. However, these two properties are substantially different in terms of the testing procedure and result obtained. The compression set is usually used to measure the polymer foam’s ability to resist permanent deformation after being subjected to an elevated temperature for up to 6 hours. The foam is subjected to a constant compressive force on a compression set device in a high-temperature oven throughout the entire test duration. On the other hand, a compression test is typically used to measure the mechanical strength of the foam by using a universal testing machine (UTM) in a uniaxial compression force loading direction towards the test sample. Compression set is used to determine how good the foam’s recovery after being subjected to a long duration of compression by constant loading (the measuring results are in the percentage of recovery, % unit), while a compression test is used to measure the basic mechanical properties of the foam including compressive stress and compressive modulus (measuring results is in MPa unit). Compression set is not measuring compressive stress and modulus. These two tests that measure the different aspects of the foam are relatively important in the shoe manufacturing industry. Therefore, these two tests were included in the characterization of our work and manuscript.      

Reviewer 3 Report

Title: Nanocomposite foams with balanced mechanical properties and energy return from EVA and MWCNT for the midsole of sports footwear application

1.      The SEM results shown for Figure 6 are unjustified. I doubt these results. The black parts should be CNTs and the white parts should be EVA. For example: the area of the black parts for 1 wt% should be ten times 0.1wt%, while it is not the case at all.

2.      What is the reason for the unusual behavior at 0.1-0.5 wt% in Figure 6? (Increase in density and decrease and increase again)

3.      The method of mixing nanoparticles with foam is very basic. How have you controlled the clumping of particles? How did you create a homogenous mixture?

4.      Results and conclusion part needs more discussion (Especially for figures 6-8).

5.      Please add more current papers in the literature and improve the introduction section. Some interesting papers related to the topic of this manuscript could be:

·         Experimental and numerical study on HDPE/SWCNT nanocomposite elastic properties considering the processing techniques effect. Microsystem Technologies 26(8), 2423-2441.

·         Experimental studies on elastic properties of high density polyethylene-multi walled carbon nanotube nanocomposites. Steel and Composite Structures, An International Journal 38 (2), 177-187.

6.      The writing of this paper needs to be improved and polished. Some clumsy and neglectful expressions can be found.

Author Response

Response to Reviewer’s Comments

Reviewer 3

  1. The SEM results shown for Figure 6 are unjustified. I doubt these results. The black parts should be CNTs and the white parts should be EVA. For example: the area of the black parts for 1 wt% should be ten times 0.1wt%, while it is not the case at all.

Thanks for the comments. The magnification of the SEM in Figure 6 is only 50X. CNTs cannot be seen under this magnification condition. The main purpose of Figure 6 in our manuscript is to show the cell size changes of the EVA foams after the incorporation of different content of CNTs, and it is not to check the CNTs in EVA. Cell size measurement and changes are one of the important parameters in determining polymer foams' performance.

  1. What is the reason for the unusual behavior at 0.1-0.5 wt% in Figure 6? (Increase in density and decrease and increase again)

Thanks for the comments. We have conducted additional measurements on the foams’ cell density counts to confirm the results and we have included the error bars /standard deviation in Figure 7. Please refer to the revised Figure 7 in our revised manuscript for details. 

  1. The method of mixing nanoparticles with foam is very basic. How have you controlled the clumping of particles? How did you create a homogenous mixture?

Thanks for the comments. The two-rolled mill is an established processing method for rubber mixing with all other ingredients. This includes high loading amount of carbon black nanoparticles and silica nanoparticles in the rubber matrix to produce various rubber-based products. Please refer to the following papers for good mixing and uniform nanoparticle dispersion using a two-rolled mill machine: 

Khan, M., Mishra, S., Ratna, D., Sonawane, S. and Shimpi, N.G., 2020. Investigation of thermal and mechanical properties of styrene–butadiene rubber nanocomposites filled with SiO2–polystyrene core–shell nanoparticles. Journal of Composite Materials, 54(14), pp.1785-1795.

DOI    https://doi.org/10.1177/0021998319886618

Wu, X., Lu, C., Zhang, X. and Zhou, Z., 2015. Conductive natural rubber/carbon black nanocomposites via cellulose nanowhisker templated assembly: tailored hierarchical structure leading to synergistic property enhancements. Journal of Materials Chemistry A, 3(25), pp.13317-13323.

DOI     https://doi.org/10.1039/C5TA02601F

The homogeneous distribution of components within the EVA elastomer is indicated by the uniform texture and color of the rolled EVA sheet after the high-speed mixing in the two-rolled mill. The details of mixing conditions and the appropriate mixing time were added in the revised manuscript accordingly. Please refer to page 3 of our revised manuscript for details.

  1. Results and conclusion part needs more discussion (Especially for figures 6-8).

Thanks for the comments. As suggested, the discussion in the results and conclusion parts (Especially for Figure 6-8) have been revised and further elaborated. Please refer to our revised manuscript for details. 

  1. Please add more current papers in the literature and improve the introduction section. Some interesting papers related to the topic of this manuscript could be:
  • Experimental and numerical study on HDPE/SWCNT nanocomposite elastic properties considering the processing techniques effect. Microsystem Technologies 26(8), 2423-2441.
  • Experimental studies on elastic properties of high density polyethylene-multi walled carbon nanotube nanocomposites. Steel and Composite Structures, An International Journal 38 (2), 177-187.

Thanks for the suggestions. We have included the suggested reference in the Introduction Section of our manuscript accordingly. Please refer to page 2, Introduction section of our revised manuscript for details.    

  1. The writing of this paper needs to be improved and polished. Some clumsy and neglectful expressions can be found.

Thanks for the comments. We have sent our manuscript for English proofreading by a professional English native speaker. Please refer to our revised manuscript for details.  

Round 2

Reviewer 3 Report

Accepted.